# Development and Evaluation of Gellan Gum/Silk Fibroin/Chondroitin Sulfate Ternary Injectable Hydrogel for Cartilage Tissue Engineering

**DOI:** 10.3390/biom11081184

**Published:** 2021-08-11

**Authors:** Seongwon Lee, Joohee Choi, Jina Youn, Younghun Lee, Wooyoup Kim, Seungho Choe, Jeongeun Song, Rui L. Reis, Gilson Khang

**Affiliations:** 1Department of Bionanotechnology and Bio-Convergence Engineering, Jeonbuk National University, 567 Baekje-daero, Deokjin-gu, Jeonju-si 54896, Korea; tjddnjs5140@jbnu.ac.kr (S.L.); zooheechoi@jbnu.ac.kr (J.C.); yjina@jbnu.ac.kr (J.Y.); lyh13256@jbnu.ac.kr (Y.L.); snail2684@jbnu.ac.kr (W.K.); tmdgh8898@jbnu.ac.kr (S.C.); songje@jbnu.ac.kr (J.S.); 23B’s Research Group, I3Bs—Research Institute on Biomaterials, Biodegradables and Biomimetics, University of Minho, Headquarters of the European Institute of Excellence on Tissue Engineering and Regenerative Medicine, AvePark, Parque de Ciência e Tecnologia, Zona Industrial da Gandra, 4805-017 Guimarães, Portugal; rgreis@i3bs.uminho.pt; 3Department of PolymerNano Science & Technology and Polymer Materials Fusion Research Center, Jeonbuk National University, 567 Baekje-daero, Deokjin-gu, Jeonju-si 54896, Korea

**Keywords:** gellan gum, silk fibroin, chondroitin sulfate, ternary hydrogel, cartilage tissue engineering

## Abstract

Hydrogel is in the spotlight as a useful biomaterial in the field of drug delivery and tissue engineering due to its similar biological properties to a native extracellular matrix (ECM). Herein, we proposed a ternary hydrogel of gellan gum (GG), silk fibroin (SF), and chondroitin sulfate (CS) as a biomaterial for cartilage tissue engineering. The hydrogels were fabricated with a facile combination of the physical and chemical crosslinking method. The purpose of this study was to find the proper content of SF and GG for the ternary matrix and confirm the applicability of the hydrogel in vitro and in vivo. The chemical and mechanical properties were measured to confirm the suitability of the hydrogel for cartilage tissue engineering. The biocompatibility of the hydrogels was investigated by analyzing the cell morphology, adhesion, proliferation, migration, and growth of articular chondrocytes-laden hydrogels. The results showed that the higher proportion of GG enhanced the mechanical properties of the hydrogel but the groups with over 0.75% of GG exhibited gelling temperatures over 40 °C, which was a harsh condition for cell encapsulation. The 0.3% GG/3.7% SF/CS and 0.5% GG/3.5% SF/CS hydrogels were chosen for the in vitro study. The cells that were encapsulated in the hydrogels did not show any abnormalities and exhibited low cytotoxicity. The biochemical properties and gene expression of the encapsulated cells exhibited positive cell growth and expression of cartilage-specific ECM and genes in the 0.5% GG/3.5% SF/CS hydrogel. Overall, the study of the GG/SF/CS ternary hydrogel with an appropriate content showed that the combination of GG, SF, and CS can synergistically promote articular cartilage defect repair and has considerable potential for application as a biomaterial in cartilage tissue engineering.

## 1. Introduction

One of the major challenges faced by developed societies today is the transition to a much older population. Due to population aging, osteoarthritis (OA) is a challenge and priority in healthcare systems around the world [1]. Severely damaged cartilage is difficult to self-recover due to its avascular, aneural, and nonlymphatic properties, and it leads to OA [2]. Currently, several surgical methods such as microfractures, bone grafts, and prosthetic joint replacements are employed to treat cartilage defects [3,4]. However, the lack of donors, demands for secondary surgery, etc. limit the application of these cartilage repair techniques [5].

The emergence of cartilage tissue engineering has led to a surge in biomaterial demand. The objective in designing biomaterials is to provide a good environment for cell growth and mechanically support regenerated tissue [6,7]. As a kind of biomaterial, hydrogel has been widely studied as an alternative to cartilage tissue due to its inherent water content, high hydrophilicity, and viscoelasticity similar to cartilage tissue [8]. Biology has shown that hydrogel can successfully fuse in the microenvironment of cartilage tissue. The hydrogel can be manipulated by the crosslinking density or crosslinking methods of polymers to optimize physical properties such as hydrophobicity, mechanical property, and biocompatibility [9,10,11]. In particular, a hydrogel can be injectable, which can avoid open surgery [12]. In addition, injectable hydrogels are excellent at filling irregular cartilage defects.

Synthetic and natural polymers are used to create a three-dimensional (3D) environment [13,14]. Synthetic polymers are easily manufactured and replicated but have poor biocompatibility and biodegradability. Nature-derived polymers such as collagen [15,16], chondroitin sulfate (CS) [17,18], hyaluronic acid (HA) [19,20], chitosan [21,22,23], silk fibroin (SF) [24], alginate [25], and gellan gum (GG) [26,27] are popularly applied in cartilage tissue engineering due to their excellent biocompatibility, biodegradability, and similar 3D microenvironment in vivo. However, the poor mechanical properties, fast degradation, and enhanced microenvironment for cell proliferation and differentiation are still challenges for practical application [28].

To date, various biomaterials for cartilage tissue engineering have been developed. However, they are still in the research stage and only a few materials have been commercialized. Therefore, it is necessary to continuously search for effective materials and develop a simple and cost-effective biomaterial for cartilage regeneration.

Herein, a composite of CS, SF, and GG was applied in designing a hydrogel for cartilage tissue engineering. CS is a significant component of the ECM of cartilage tissue [29]. The polysaccharide units of CS are composed of carboxylic acids and sulfate ester groups, resulting in a high negative polarity, which adsorb and concentrate positively charged growth factors to induce cell adhesion, differentiation, and migration. However, due to the low molecular weight and poor mechanical properties, pristine CS has limitations in the application in cartilage tissue engineering [22,30]. SF is a naturally occurring protein polymer that has been used in the production of clinical sutures for centuries. SF is popularly applied in tissue engineering due to its effective cellular compatibility. The application of SF in cartilage tissue engineering has been shown as a promising biomaterial and has shown superior support in the adhesion, differentiation, and proliferation of chondrocytes. However, a single component of SF has a slow gelation rate, which restricts applicability in the biomedical field. Physical methods such as external stimuli via sonication, vortexing, etc., or changing the pH or temperature may induce the gelation of SF but are difficult to handle [17,18,24]. GG is actively used in various fields such as drug delivery systems and tissue engineering [26,31]. GG is a negatively charged linear polysaccharide consisting of four repeating units, one L-rhamnose, D-glucuronic acid, and two D-glucose carbohydrates [32]. The gelation process of GG occurs physically by forming random coils at high temperatures and forming double helices at low temperatures [33]. The GG can also be crosslinked through ionic crosslinking with the existence of cations such as Ca^2+^, Mg^2+^, Na^+^, and K^+^ [34]. The attractive advantage of the GG for cartilage tissue engineering is the efficient delivery of chondrocytes and support of the deposition of cartilage-specific ECM by providing a microenvironment similar to native glycosaminoglycans from the existence of glucuronic acid residues in the repeat unit [35]. In addition, the mechanical properties of the GG are easy to control by changing the concentration or ionic components, and they show similar elastic moduli to soft tissue [34,35]. However, the GG lacks a cell-adhesive site, which limits the providing of a proper microenvironment for the migration, proliferation, and differentiation of cells [32,33].

We hypothesized that the ternary matrix of CS, SF, and GG may complement the shortcomings of each biomaterial and integrate the merits to construct cartilage tissue engineering biomaterial. The CS provided a similar microenvironment to that of native cartilage. The SF supports cell attachment and migration. The GG was applied to improve overall mechanical properties. The hydrogel was fabricated by varying the SF and GG ratio and CS was fixed in all the groups. The SF was chemically crosslinked through carbodiimide chemistry and GG was physically crosslinked through ionic crosslinking. Physicochemical properties and mechanical properties were analyzed, and the proper content of SF and GG was selected and applied in an in vitro study to characterize the biological properties of primary chondrocytes that are encapsulated in the hydrogels.

## 2. Materials and Methods

### 2.1. Preparation of SF Solution

First, 10 g of silkworm cocoons from *bombyx mori* (Kyebong Farm, Cheongyang, Korea) were boiled in 0.02 M of sodium carbonate (Na_2_CO_3_, Showa Chemical, Tokyo, Japan) that was dissolved in distilled water (DW) at 100 °C for 30 min to remove the sericin. The degummed silkworm cocoons were rinsed with DW 3 times and were dried in a 60 °C oven. After the degummed silkworm cocoons were fully dried, 7 g of the dried silk was dissolved in 9.3 M of lithium bromide monohydrate (LiBr, Kanto Chemical Co., Tokyo, INC., Japan) at 60 °C for 4 h. Then, it was dialyzed in DW using a SnakeSkin^Ⓡ^ dialysis tubing (molecular weight cut-off: 3500, Thermo Fisher Scientific, Waltham, MA, USA) for 72 h to remove the residual LiBr. The final concentration of the SF solution was 8% (*w*/*v*) and the SF solution was stored at 4 °C until further use.

### 2.2. Fabrication of Hydrogels

Low-acyl gellan gum (GG; Gelzan TM CM, Sigma-Aldrich, St. Louis, MO, USA) was added in 80 °C DW with an amount of 0.3%, 0.5%, 0.75%, and 1% (*w*/*v*) each and was stirred until it was fully dissolved. Then, 10 mM of calcium chloride (CaCl_2_, Sigma-Aldrich, USA) was added to the GG solution for the ionic crosslinking. Then, CS from the bovine trachea (Sigma-Aldrich, USA) was added to each GG solution with an amount of 1.5% (*w*/*v*) and stirred until it was fully dissolved. After CS was evenly dissolved in the GG solution, the temperature was lowered to 50 °C. SF solution was chemically crosslinked with 1-ethyl-3-(3-dimethyl aminopropyl) carbodiimide hydrochloride (EDC, Sigma-Aldrich, USA) and 1-ethyl-N-hydroxysuccinimide (NHS, Sigma-Aldrich, USA) with an amount of 20 wt.% and 10 wt.% of SF mass, respectively [36]. The crosslinked SF solution was included in each GG/CS solution to make the total concentrations 3.7%, 3.5%, 3.25%, and 3% of SF, which are specified as 0.3% GG/3.7% SF/CS, 0.5% GG/3.5% SF/CS, 0.75% GG/3.25% SF/CS, and 1% GG/3% SF/CS, respectively. The SF blended solutions were stirred for 30 min, and 7 mL of the solutions was poured on a 35 mm petri dish and stored at room temperature (RT) for gelation. The solidified samples were punched with a 6 mm biopsy punch (Kai Medical, Seki, Japan) for characterization. 

### 2.3. Fourier-Transform Infrared (FT-IR) Spectroscopy

The chemical properties of the fabricated hydrogels were characterized using an attenuated total reflectance-Fourier-transform infrared spectrometer (ATR-FTIR, Perkin Elmer, Boston, MA, USA) in the spectral range of 400–4000 cm^−1^. The resolution was 0.4–64 cm^−1^ and the crystal type was diamond/KRS-5 (thallium bromoiodide) crystal. All the hydrogels were freeze-dried for the measurement.

### 2.4. Physicochemical Property Characterization

#### 2.4.1. Morphological Observation

The hydrogels were frozen in a deep freezer (−80 °C) and lyophilized for 72 h to remove water. The lyophilized hydrogels were cross-sectioned and placed on a carbon-taped grid and coated by a plasma sputter (Model SC500k, Emscope, UK) for analysis. The porous structure was observed by a Bio-LV scanning electron microscope (Bio-LV SEM, Japan, HITACHI). The pore size of the hydrogels was analyzed by using ImageJ software.

#### 2.4.2. Swelling Ratio (%)

The swelling ratio (%) of the hydrogels was characterized at 0.5, 1, 2, 3, 4, 8, and 24 h of incubation. The initial weight of the hydrogels (*W_i_*) was recorded, and the samples were immersed in phosphate-buffered solution (PBS, Thermo Fisher Scientific, USA) solution and were stored in a 37 °C incubator. At the specific time point, the supernatant was removed and the weight of the swollen hydrogels (*W_s_*) was measured. The swelling ratio (%) was calculated using formula (1).
(1)$$\mathrm{Swelling}\;\mathrm{ratio}\;\left(\%\right)=\frac{W_{s}-W_{i}}{W_{s}}\times 100\;(\%)$$

#### 2.4.3. Degradation Ratio (%)

The degradation ratio (%) of the hydrogels was characterized on 1, 3, 7, 14, 21, and 28 days of culture in trypsin (0.04% in PBS, Thermo Fisher Scientific, USA). All the samples were stored in a 37 °C incubator for 24 h to obtain fully swollen samples. The weight of the hydrogels was recorded (*W_w_*) as an initial weight. At a specific time point, the supernatant was removed and the weight of the samples was measured (*W_t_*). The solution was changed every 3 days to a fresh one. The degradation ratio (%) was calculated using formula (2).
(2)$$\mathrm{Degradation}\;\mathrm{ratio}\;\left(\%\right)=\frac{W_{w}-W_{t}}{W_{w}}\times 100\;(\%)$$


#### 2.4.4. Sol Fraction (%)

The sol fraction of the hydrogels was measured by freeze-drying the fabricated hydrogels. The initial weight of the dried samples was weighed (*W_d_*). Each specimen was incubated in 1 mL of DW with slight agitation (60 rpm) for 1 h. Then, DW was removed and lyophilized for 48 h. The remaining weight of the samples was measured (*W_r_*) and the sol fraction (%) was calculated using formula (3).
(3)$$\mathrm{Sol}\;\mathrm{fraction}\;\left(\%\right)\;=\frac{W_{d}-W_{r}}{W_{d}}\times 100\;\left(\%\right)$$

### 2.5. Mechanical Characterization

#### 2.5.1. Compression Test

The compressive modulus was measured by a texture analyzer (FTC, Sterling, Virginia, USA). The hydrogels 4 mm in height and 6 mm in diameter were prepared. The test was performed under an unconfined condition with a speed of 2 mm/min and a load cell of 10 N. The initial elastic modulus was calculated at 10–15% strain.

#### 2.5.2. Extrusion Force Analysis

The extrusion force was carried out by following the previous reported study with a slight modification [37]. The hydrogel solutions were manufactured and loaded in a 1 mL syringe (Kovax-syringe, Korea Vaccine Co., Ltd, Songpa-gu, Korea) and capped with a 22-gauge needle. The samples were stored at RT for 5 min and were extruded with a texture analyzer with a speed of 20 mm/min and a load cell of 10 N.

#### 2.5.3. Viscosity Test

The viscosity of the hydrogels and the gelation temperature were measured with a viscometer (Ametek Brookfield, Middleborough, MA, USA). The circulation tank was preheated to 50 °C, and 8 mL of a pre-made hydrogel solution was added to a viscometer. The router speed was set to 1 rpm and the cone and plate spindle (SC-32 spindle, AMETEK Brookfield, USA) was used in this study. The temperature was gradually lowered until it reached 20 °C.

### 2.6. In Vitro Study

#### 2.6.1. Isolation of Chondrocyte from Articular Cartilage of Rabbit Knee

All animal experiments were performed following the guidelines and approval of Jeonbuk National University Animal Care Committee, Jeonju, Republic of Korea (JBNU 2016-50). Before the procedure, all surgical instruments were sterilized in an autoclave and cell isolation was performed under a clean bench. Chondrocytes were isolated from New Zealand white rabbits (6 weeks, female, Hanil Scientific, Gimpo, Korea). Rabbit legs were collected and were washed three times with 1% penicillin-streptomycin (PS, Thermo Fisher Scientific, USA) under the clean bench. The leg bones were cut off and the soft tissue surrounding the cartilage of the thigh bones was removed. Using a surgical blade, cartilage tissue was peeled and collected. The obtained tissue was centrifuged under conditions of 1200 rpm, 4 °C, and 3 min. The supernatant was removed and the collected tissue was digested in collagenase A (Roche Diagnostics International Ltd., Basel, Switzerland) overnight to obtain chondrocytes. The digested solution was centrifuged under conditions of 1200 rpm, 4 °C, and 3 min and the collected chondrocytes were cultured in cell culture dishes (Eppendorf, Hamburg, Germany). The cells were cultured in Dulbecco’s modified Eagle’s medium: nutrient mixture F-12 (DMEM/F-12, Thermo Fisher Scientific, USA) cell culture medium that contains 10% fetal bovine serum (FBS, Thermo Fisher Scientific, USA) and 1% PS. The cell culture medium was exchanged for a fresh one every 3 days until the cells reached passage 2.

#### 2.6.2. Cell Encapsulation in Hydrogels

All the experimental tools and materials were sterilized in an autoclave before producing a cell-laden hydrogel. Chondrocytes in cell culture dishes were collected by trypsinizing with trypsin-EDTA (Thermo Fisher Scientific, USA). Then, 1 × 10^6^ cells/mL of the chondrocytes were mixed with the hydrogel solution. The cell-laden hydrogel solution was poured into a petri dish and punched with a 6 mm punch after gelation occurred. The manufactured hydrogel (6 mm diameter and 3 mm height) was transferred to a 24-well plate using a sterilized spatula. The cell-laden hydrogels were incubated in the DMEM/F12 cell culture medium and were changed every 3 days until they reached specific days.

#### 2.6.3. Live and Dead Staining

The live and dead assay was carried out to study cell proliferation and viability [38]. The samples were evaluated by using a live and dead cell imaging kit (Invitrogen, Carlsbad, CA, USA). The hydrogels were washed with PBS 3 times and were transferred to a confocal dish (Coverglass bottom dish, SPL Lifesciences, Pocheon-si, Korea). The hydrogels were stained with calcein AM and ethidium homodimer and incubated in the cell culture incubator (5% CO_2_ and 37 °C) for 30 min. The cell images were obtained using a super-resolution conformal laser scanning microscope (LSM 880, Airyscan, Carl Zeiss, Jena, Germany) along with the Z-stack process.

#### 2.6.4. GAG and dsDNA Quantitative Analysis

The cell-laden hydrogels were washed with PBS 3 times at specific time points and the hydrogels were stored in a deep freezer (−60 °C) until all the samples were prepared. Cell lysis and digestion were carried out in papain solution, which was fabricated by dissolving 0.0875 g of L-cysteine (Sigma-Aldrich, USA) and 6.25 mg of papain powder (Sigma-Aldrich, USA) in PBE solution, which was prepared by adding 0.71 g of sodium phosphate dibasic heptahydrate (Na_2_HPO_4_, Sigma-Aldrich, USA) and 0.186 g of ethylene diamine tetraacetic acid trisodium salt hydrate (EDTA, Sigma-Aldrich, USA) in 50 mL of DW. Each cell-laden hydrogel was loaded in the fabricated papain solution and was stored in a 60 °C oven. The extracted solution was stored at −20 °C until the further experiment. The GAG content analysis was carried out by following the standard 1,9-dimethyl methylene blue (DMMB) staining. Briefly, the DMMB solution reagent was fabricated by dissolving 1 g of sodium formate (Sigma-Aldrich, USA) and 8 mg of DMMB powder (Sigma-Aldrich, USA) in 500 mL of DW. Next, 2.5 mL of 100% ethanol and 1 mL of formic acid (Sigma-Aldrich, USA) were added. The pH of the solution was adjusted to 3.5 by adding 0.1 M of acetic acid (Sigma-Aldrich, USA). Then, 20 μL of the digested samples and 200 μL of the fabricated DMMB solution was added to a 96-well plate. The GAG content was analyzed at a wavelength of 525 nm using a microplate reader. The concentration was calculated by using chondroitin sulfate as a standard curve. dsDNA content was analyzed by using Quant-iT PicoGreen reagent (Life Technologies, Carlsbad, CA, USA) following the manufacturer’s protocol. Briefly, the extracted solution was diluted in MES buffer, and the solution was transferred into a 96-well black plate and Quanti-iT PicoGreen reagent was included in the samples. The samples were incubated at RT for 5 min and the dsDNA content was analyzed with a microplate reader (Emax Molecular Devices, USA) at an excitation wavelength of 485/20 nm and an emission wavelength of 528/20 nm. The concentrations of 0 μg/mL to 2 μg/mL were applied as a standard curve.

#### 2.6.5. Morphology and Histological Observation

The cell-encapsulated hydrogels were cultured for 3 and 21 days and were washed with PBS 3 times. The samples were treated with 2.5% glutaraldehyde and stored at 4 °C for 24 h. After 24 h, the hydrogels were rinsed with PBS 3 times to remove residual glutaraldehyde. The SEM observation was carried out following the method in Section 2.4.1. For histological observation, the fixed samples were frozen with a Cryomatrix. All the samples were sectioned into 10 µm thickness with a cryomicrotome (Thermo Fisher Scientific, USA). Hematoxylin & Eosin (H&E), Safranin-O (SO), Toluidine Blue (TB), and Alcian Blue (AB) staining were carried out following the standard histological techniques.

#### 2.6.6. Real-Time Polymerase Chain Reaction (RT-PCR)

RT-PCR was performed to examine the gene expression of the cell-encapsulated hydrogels [39]. The samples were cultured in the DMEM/F-12 for 3 and 21 days. The samples were washed with PBS 3 times. The cell lysis of the samples was performed by homogenizing with Trizol (Takara Bio Inc., Shiga, Japan) using a glass tissue grinder (Weaton Industries, Millville, NJ, USA). The homogenized solutions were transferred to a 1.5 mL Eppendorf tube (EP tube, Eppendorf, Germany) and chloroform (Samchun chemicals, Gangnam-gu, Korea) was included. The samples were centrifuged at 12,000× *g* and 4 °C for 15 min. The supernatant was transferred to a 1.5 mL EP tube and isopropanol was added and stored at 4 °C for 4 h. After 4 h, the samples were centrifuged at 12,000× *g* and 4 °C for 15 min. The supernatant was removed and 75% of ice-cold ethanol was added and centrifuged at 7000× *g* and 4 °C for 5 min. The extracted mRNA was diluted with free water (UltraPure^TM^ Distilled Water, Life technologies, USA). The mRNA concentration was analyzed using the BioSpectrophotometer (Eppendorf, Germany) and amplification was performed with a PCR Thermal Cycler (Takara Bio Inc., Japan). The RT-PCR was carried out with an SYBR™ Green PCR Master Mix (Applied Biosystems, Middlesex, MA, USA) and StepOnePlus Real-Time PCR system (Applied Biosystems, USA). All genes were normalized by the housekeeping gene GAPDH.

### 2.7. Statistics

All the numerical results are presented by the mean ± standard deviation (SD). The GraphPad Prism 5.0 software (GraphPad Software, La Jolla, CA, USA) was utilized to perform the statistical analysis. The studies were analyzed employing a one-way analysis of variance (one-way ANOVA test), and the differences were considered significant at *p* < 0.05(*), *p* < 0.01(**), and *p* < 0.001(***). 

## 3. Results and Discussion

### 3.1. Characterization of Hydrogels

#### 3.1.1. FT-IR Analysis

FT-IR analysis was performed to identify interactions between GG, crosslinked GG, SF, crosslinked SF, CS, and mixture hydrogels (Figure 1). The specific peaks of the GG were detected at 3300 cm^−1^ (-OH stretching group), 1596 cm^−1^ (C-H_2_ stretching vibration group), and 1035 cm^−1^ (–COC stretching group) [40]. CS represented a specific peak similar to GG [41]. The pristine SF exhibited characteristic peaks at 3300 cm^−1^ (–NH stretching group), 1636 cm^−1^ (amide I), 1556 cm^−1^ (amide II), and 1232 cm^−1^ (amide III) [42]. SF crosslinked with EDC-NHS (ENSF) showed a strong peak at 1716 cm^−1^, an amide group that confirms chemical crosslinking between the carboxyl group of SF and the amine group of EDC [43]. The 0.3% GG/3.7% SF/CS, 0.5% GG/3.5% SF/CS, 0.75% GG/3.25% SF/CS, and 1% GG/3% SF/CS hydrogels displayed characteristic peaks at 3302 cm^−1^, 3302 cm^−1^, 3306 cm^−1^, and 3303 cm^−1^ (–OH and -NH stretching); 1634 cm^−1^, 1634 cm^−1^, 1633 cm^−1^, and 1634 cm^−1^ (amide I); 1557 cm^−1^, 1557 cm^−1^, 1563 cm^−1^, and 1563 cm^−1^ (amide II); 1221 cm^−1^, 1227 cm^−1^, 1229 cm^−1^, and 1226 cm^−1^ (amide III); and 1026 cm^−1^, 1030 cm^−1^, 1029 cm^−1^, and 1029 cm^−1^ (–COC stretching). The shifting of the peak may be due to the intermolecular interactions of GG, SF, and CS [44]. Besides the characteristic peaks corresponding to each matrix, all hydrogels did not show any new peaks, which shows that the interaction among the ternary matrix was formed by physical interaction [44].

#### 3.1.2. Morphological Observation of Hydrogel

The pore size of the scaffold and interconnectivity of the microenvironment is undoubtedly an important factor in tissue engineering applications [45]. All the groups displayed porous structures (Figure 2a). The pore sizes of the 0.3% GG/3.7% SF/CS, 0.5% GG/3.5% SF/CS, 0.75% GG/3.25% SF/CS, and 1% GG/3% SF/CS were 28.8 ± 4.53 μm, 31.2 ± 2.27 μm, 36.8 ± 3.97 μm, and 37.7 ± 5.69 μm, respectively (Figure 2b). The reason for this may be due to the lower content of chemically crosslinked SF reducing the interconnectivity of the matrix, leading to a higher pore size.

#### 3.1.3. Physicochemical Study of GG/SF/CS

Swelling is one of the important factors considered in designing biomaterials. The rate of expansion of hydrogels is related to absorbing nutrients from cell culture serum in vitro and preserving fluids in the body. It is associated with cell proliferation and ECM molecular diffusion over time and requires a swelling evaluation of hydrogel [46,47]. The swelling ratio of the hydrogels was measured for 0, 0.5, 1, 2, 3, 4, 8, and 24 h (Figure 3a). All the hydrogels showed a rapid increase in swelling at 0.5 h and then remained constant, which shows that hydrogel can reach a fully swollen state at 0.5 h. According to Zhou et al., they applied an SF and CS scaffold for cartilage regeneration and showed that the scaffold only reached ~6% of the swelling ratio [18]. In addition, Yan et al. designed a ternary scaffold by applying SF, CS, and HA and showed that the maximum swelling ratio was ~25% [17]. Compared to these results, the swelling ratio of the GG/SF/CS hydrogels was ~30% of the swelling ratio, which suggests that the GG/SF/CS hydrogel can absorb a large amount of water and nutrients for cell growth [48].

Degradation is also an important factor in designing a proper scaffold. The scaffold is important to sufficiently support cells under a microenvironment while properly degrading for cell growth and tissue regeneration [49,50]. Notably, 0.3% GG/3.7% SF/CS, 0.5% GG/3.5% SF/CS, 0.75% GG/3.25% SF/CS, and 1% GG/3% SF/CS showed a sharp increase in degradation ratio after the 14 days (Figure 3b). In particular, 0.3% GG/3.7% SF/CS and 0.5% GG/3.5% SF/CS groups showed a faster degradation after 14 days than other groups did. It is suspected that the groups with a higher amount of GG delayed the degradation from the formation of ionic crosslinking and strong hydrogen bonding. It is still a challenge to seek a suitable degradation rate for cartilage regeneration, but based on the subsequent in vitro results, in which the hydrogels did not hinder cell growth over time, it is believed that this degradation rate is appropriate for application as a cartilage tissue engineering biomaterial.

The crosslinking density was confirmed by the sol fraction, which displays a matrix that is not included in the hydrogel network [46]. Here, 0.3% GG/3.7% SF/CS, 0.5% GG/3.5% SF/CS, 0.75% GG/3.25% SF/CS, and 1% GG/3% SF/CS showed 67.5 ± 8.74%, 64.1 ± 0.78%, 59.0 ± 6.76%, and 46.9 ± 4.97%, respectively (Figure 3c). The sol fraction represents the proportion of polymer not included in the crosslinked matrix. Therefore, the higher the sol fraction, the lower the crosslinking density of the hydrogel. It can be seen that the existence of a higher amount of GG is an important factor for the overall matrix to sufficiently interact [51,52].

#### 3.1.4. Mechanical Properties Characterization

The compressive strength of the hydrogels is an essential property for cartilage tissue engineering biomaterials to support external forces. A strain rate of 2%/s was applied in this study as the physiological strain of the cartilage tissue under the loading environment is reported to be 10–20% [53]. The initial elastic moduli of 0.3% GG/3.7% SF/CS, 0.5% GG/3.5% SF/CS, 0.75% GG/3.25% SF/CS, and 1% GG/3% SF/CS showed 29.8 ± 7.75 kPa, 160.0 ± 120.2 kPa, 407.0 ± 215.66 kPa, and 630.0 ± 162.63 kPa, respectively. The result showed that the mechanical property improved as the GG content increased, which may be due to strong hydrogen bonding among the water and double-helical structure of the GG backbone. In addition, ionic crosslinking may have induced higher mechanical properties (Figure 4a). It is known that the results of the elastic modulus for SF-based hydrogels studied in previous studies are mainly 369–1712 kPa, which suggests that GG/SF/CS can be considered to have sufficient mechanical properties [54,55]. In addition, according to a previous study of a physically modified GG-based hydrogel, the peak load of the composite hydrogel only reached 3.5–5.3 kPa in HA-blended GG and pullulan-blended GG hydrogel [56,57]. Moreover, the compression strength of chitosan-loaded GG only reached 10–20 kPa, and gelatin and chitosan-loaded GG showed 10–25 kPa from a compression test. Based on these results, it can be regarded that the GG/SF/CS has significantly enhanced the mechanical property of the GG-based composite. The s–s curve of the 0.75% GG/3.25% SF/CS and 1% GG/3% SF/CS hydrogels exhibited a significantly higher ultimate stress as the strain increased (represented as blue and red arrows) and showed significantly higher strength (Figure 4b). This is due to the high interaction among the ternary matrix and existence of higher physical crosslinking of GG matrix. 

The injection force was measured to analyze the injectability of the hydrogels. The injection forces of each hydrogel were 1.1 N, 2.2 N, 3.0 N, and 4.3 N in 0.3% GG/3.7% SF/CS, 0.5% GG/3.5% SF/CS, 0.75% GG/3.25% SF/CS, and 1% GG/3% SF/CS, respectively (Figure 4c). The extrusion force of the hydrogel increased with the GG content in the hydrogel composite. This may be due to the high molecular weight and ionic crosslinking density of the GG interfering with the injectability of the hydrogel. On the other hand, groups where the GG concentration lowered and SF content increased exhibited a shear-thinning property even though SF was chemically crosslinked, indicating that this hydrogel can be applied as an injectable hydrogel.

The gelation temperature and viscosity of the hydrogels were characterized according to the temperature change. The gelation needs to occur at an appropriate temperature that does not affect cell survival and allows a uniform suspension of cells in the hydrogels [58,59]. The gelation temperatures of each hydrogel were 33.7 °C, 38.7 °C, 41.4 °C, and 45.6 °C in 0.3% GG/3.7% SF/CS, 0.5% GG/3.5% SF/CS, 0.75% GG/3.25% SF/CS, and 1% GG/3% SF/CS, respectively (Figure 4d). The storage modulus of the hydrogels after the gelation occurred showed higher values in the groups with the higher amount of GG hydrogels, which showed a similar tendency of the compression test. The overall results show that the mechanical properties of the hydrogels were not significantly affected by the content of the SF but were rather affected by the GG. Therefore, the overall mechanical properties of the hydrogel can be adjusted according to the GG concentration and ionic crosslinking density. The biological properties of the hydrogels were analyzed with 0.3% GG/3.7% SF/CS and 0.5% GG/3.5% SF/CS as the gelation temperature was proper for cells encapsulation.

### 3.2. In Vitro Analysis

#### 3.2.1. Live/Dead Staining

Creating an environment in which chondrocytes can survive and grow for cartilage tissue reconstruction is one of the most important factors. The biocompatibility of the produced hydrogel was analyzed by encapsulating chondrocytes in the hydrogel. The live/dead dyeing reagents calcein-AM (green) and ethidium homodimer (red) dyed live and dead cells, respectively (Figure 5a). On 3 days of culture, all hydrogels were found to be well attached to the surface and only a few cells died (Figure 5b). Of these, 0.3% GG/3.7% SF/CS was observed in more cells than 0.5% GG/3.5% SF/CS. This means that 0.3% GG/3.7% SF/CS has a small pore size, which is why more cells are attached to it. However, there was more cell proliferation at 0.5% GG/3.5% SF/CS during 21 days of incubation. This provides an advantage over 0.5% GG/3.7% SF/CS in cellular adhesion and proliferation. According to the previous study, Kim et al. encapsulated chondrocytes in HA-loaded gellan gum and showed the highest viability of GG hydrogel on 3 days of culture [56]. However, on 21 days of culture, it was found that the number of cells that died due to the lack of cell attachment was high. The integration of SF improved cell adhesion and proliferation, and the integration of CS facilitated cell migration [54,60]. This is considered to have great potential as a hydrogel in cartilage tissue engineering because GG/SF/CS is suitable for cells and provides an advantageous environment for cell attachment and proliferation.

#### 3.2.2. dsDNA and GAG

Quantitative analysis of dsDNA for cell proliferation was evaluated on 3, 7, 14, and 21 days of culture (Figure 6a). The dsDNA content on 3 days of culture was 0.4 ± 0.15 μg/mL and 0.4 ± 0.07 μg/mL in 0.3% GG/3.7% SF/CS and 0.5% GG/3.5% SF/CS, respectively. The dsDNA content of both groups decreased on 7 days of culture, possibly due to the decomposition of the hydrogels. On 14 days of culture, the dsDNA content of 0.5% GG/3.5% SF/CS showed a higher amount when compared to 7 days of culture, which may be due to a faster cell proliferation rate than the decomposition rate of the matrix. On the other hand, the dsDNA content of the 0.3% GG/3.7% SF/CS showed a lower concentration when compared to 7 days, which may be due to the faster degradation rate than the cell proliferation rate. On 21 days of culture, all the groups showed an increased dsDNA content and, especially, the 0.5% GG/3.5% SF/CS group showed the highest dsDNA content. It is regarded that, in addition to the cell attachment site, physical support is significant for cells to proliferate in the hydrogel microenvironment [61].

GAG is an important ECM component of cartilage that provides a microenvironment suitable for chondrocytes [62]. GAG showed that GG/SF/CS hydrogels with different content ratios have different abilities to generate cartilage ECMs (Figure 6b). The value of GAG was normalized to the corresponding total dsDNA content of each sample. In addition, 0.5% GG/3.5% SF/CS had a lower dsDNA content on 3 and 7 days compared to 0.3% GG/3.7% SF/CS, but significantly higher GAG content. This may be due to the higher mechanical properties of the hydrogel, which provided an appropriate environment for chondrocyte growth. On 7 days of culture, 0.3% GG/3.7% SF/CS and 0.5% GG/3.5% SF/CS showed a lower content of GAG when compared to 3 days of culture, which is due to the degradation of the matrix. On 14 and 21 days of culture, the GAG production showed a similar pattern to the dsDNA, which may be due to the activation of cell proliferation and growth from the proper microenvironment of the hydrogels. The overall results show that the proper content ratio between GG, SF, and CS improves the proliferation and formation of cartilage ECM.

#### 3.2.3. Morphology and Histological Observation

ECM formation and cell morphology of the cell-laden hydrogels incubated on 3 and 21 days were observed by bio-LV SEM and histological analysis (Figure 7a,b). The encapsulated cells in both the experimental groups were uniformly dispersed in the porous structure and did not display any abnormal cells. The morphology of the cells was spherical, which is a favorable structure for chondrogenesis [7]. A greater number of cells were observed on 21 days of culture in both groups, which may be due to the proliferation of cells in the matrix. The histology of H&E, SO, AB, and TB was observed with the cell-laden hydrogels that were cultured for 3 and 21 days. H&E staining shows the composition of newly formed tissues and identifies the cytoplasm secreted around the nucleus and cells, and SO, AB, and TB stain the cartilage-specific matrices [63,64,65]. The staining showed well-distributed cells in the matrix and some cells showed a noticeable matrix around the cell. In particular, 0.5% GG/3.5% SF/CS hydrogel showed more chondrocytes than 0.3% GG/3.7% SF/CS hydrogel did on 21 days of culture. It can be concluded that cell attachment and proliferation occurred efficiently in the microenvironment.

#### 3.2.4. Gene Expression

For the recovery of cartilage tissue in vivo, it is important to maintain the phenotype of chondrocytes because substrate-producing cells in the cartilage contribute to the formation of new tissues [18]. Cartilage-specific genes (SRY-Box Transcription Factor 9 (SOX9), Type-II collagen (COL2), and Aggrecan (AGG)) were evaluated on 3 and 21 days of culture, and GAPDH was used as a housekeeping gene (Figure 8). AGG plays an important role in cartilage function [65,66], COL2 is a major component of cartilage tissue [67,68], and SOX9 is one of the significant transcription factors for the differentiation and formation of chondrocytes [69]. On 3 days of culture, AGG, COL2, and SOX9 expressions were 1.8-, 1.7-, and 1.7-fold higher in 0.5% GG/3.5% SF/CS compared to 0.3% GG/3.7% SF/CS, respectively. On 21 days of culture, the relative gene expressions of AGG, COL2, and SOX9 in 0.5% GG/3.5% SF/CS were 2.4-, 2.3-, and 1.8-fold higher compared to 0.3% GG/3.7% SF/CS (Figure 8), respectively. These results show that the 0.5% GG/3.5% SF/CS hydrogel can effectively maintain chondrocytes in substrate-generated phenotypes. This result may be due to the higher mechanical properties of 0.5% GG/3.5% SF/CS sufficiently supporting the cells in the matrix to proliferate and differentiate.

## 4. Conclusions

In this work, the ternary hydrogel of GG/SF/CS was proposed as an effective biomaterial for cartilage TE. Various amounts of hydrogel were manufactured in a simple method under pH-neutral conditions. The crosslinking method, which was performed by using Ca^2+^ and carbodiimide reaction, was cell-friendly. GG served as a bridgehead for overall support and SF and CS provided a similar microenvironment to that of native cartilage tissue. The physicochemical and mechanical properties of hydrogels improved with the integration of GG/SF/CS. The in vitro study with primary chondrocytes exhibited biocompatibility of the hydrogels. The biochemical properties of the cells were confirmed by dsDNA, GAG, morphology, histology, and gene expression assay. The biological activity of the chondrocytes was active in 0.5 GG/3.5% SF/CS. In conclusion, it is expected that GG/SF/CS ternary hydrogel can be applied as a promising cartilage tissue engineering biomaterial.

## Figures and Tables

**Figure 1 biomolecules-11-01184-f001:**
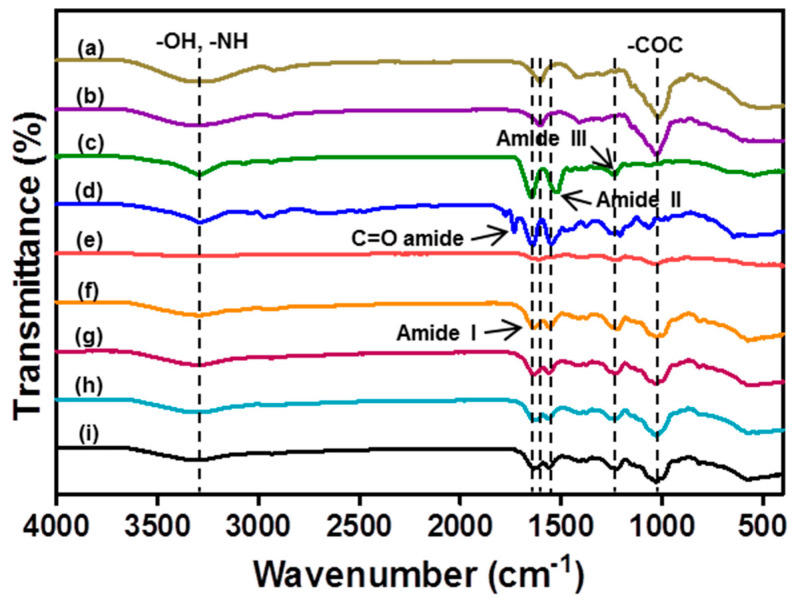
FT-IR evaluation of GG (a), ionic crosslinked GG (b), pristine SF (c), EN-SF (d), CS (e), 0.3% GG/3.7% SF/CS (f), 0.5% GG/3.5% SF/CS (g), 0.75% GG/3.25% SF/CS (h), and 1% GG/3% SF/CS (i) analyzed at the wavelength range of 400–4000 cm^−1^.

**Figure 2 biomolecules-11-01184-f002:**
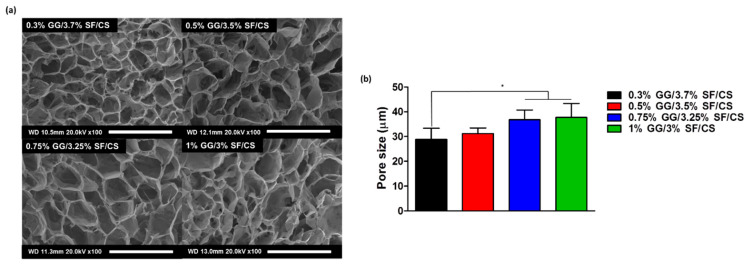
Morphological observation of the hydrogels: (**a**) SEM observation of hydrogel morphology and (**b**) pore size of hydrogel (scale bar = 100 μm) (values are mean ± SD, *n* = 3, *p* < 0.05 (*)).

**Figure 3 biomolecules-11-01184-f003:**
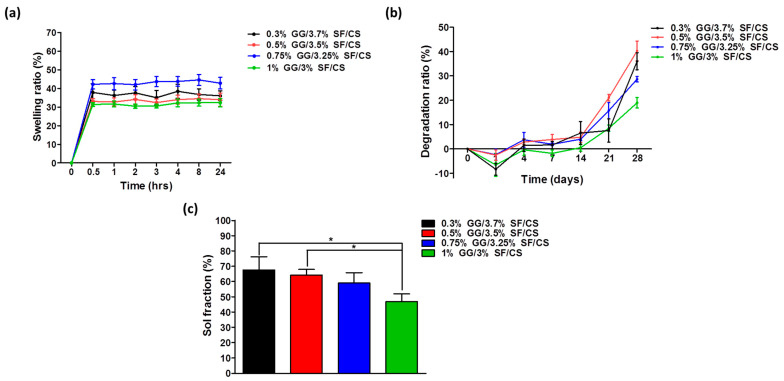
Physicochemical study of the hydrogels: (**a**) swelling ratio (%), (**b**) degradation ratio analyzed for 28 days, and (**c**) sol fraction ratio (%) (values are mean ± SD, *n* = 6, *p* < 0.05 (*)).

**Figure 4 biomolecules-11-01184-f004:**
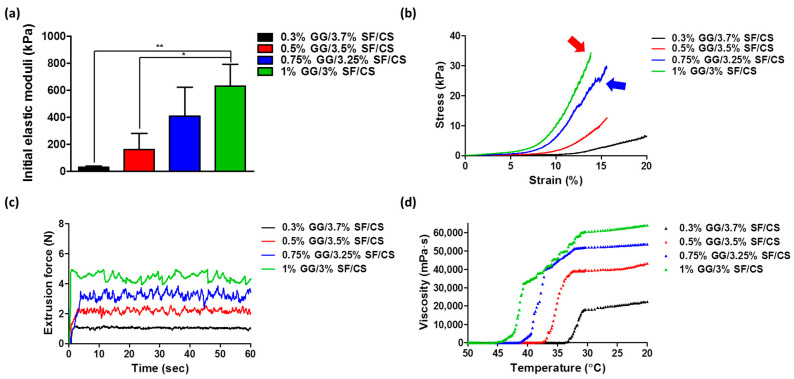
Mechanical characterization of the hydrogels: (**a**) initial elastic modulus at 5–10% strain; (**b**) stress–strain curve (s–s curve) of the hydrogels analyzed at the speed of 2 mm/min with a load cell of 10 N; (**c**) extrusion force analyzed using a 1 mL syringe capped with a needle, texture analyzer at a speed of 20 mm/min, and load cell of 20 N; and (**d**) viscosity analyzed at the temperature range from 50 °C to 20 °C (values are mean ± SD, *n* = 5, *p* < 0.05 (*) and *p* < 0.01 (**)).

**Figure 5 biomolecules-11-01184-f005:**
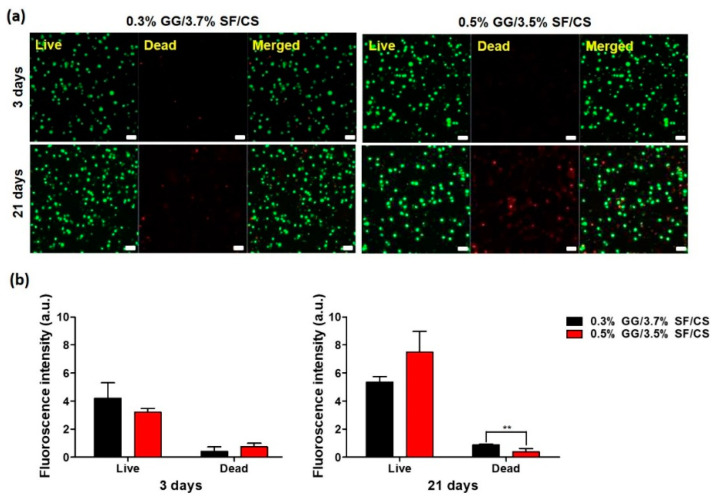
Cytotoxicity analysis of the cell-laden hydrogels: (**a**) live/dead staining images of chondrocytes-encapsulated hydrogels analyzed with the Z-stack mode, and (**b**) relative intensity of live/dead images evaluated with ImageJ software (scale bar = 50 μm) (values are mean ± SD, *n* = 3, *p* < 0.01 (**)).

**Figure 6 biomolecules-11-01184-f006:**
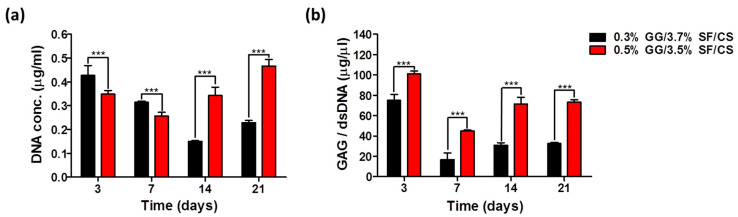
Biochemical characterization of the cell-encapsulated hydrogels: (**a**) dsDNA content and (**b**) GAG quantitative analysis (values are mean ± SD, *n* = 6, *p* < 0.001 (***)).

**Figure 7 biomolecules-11-01184-f007:**
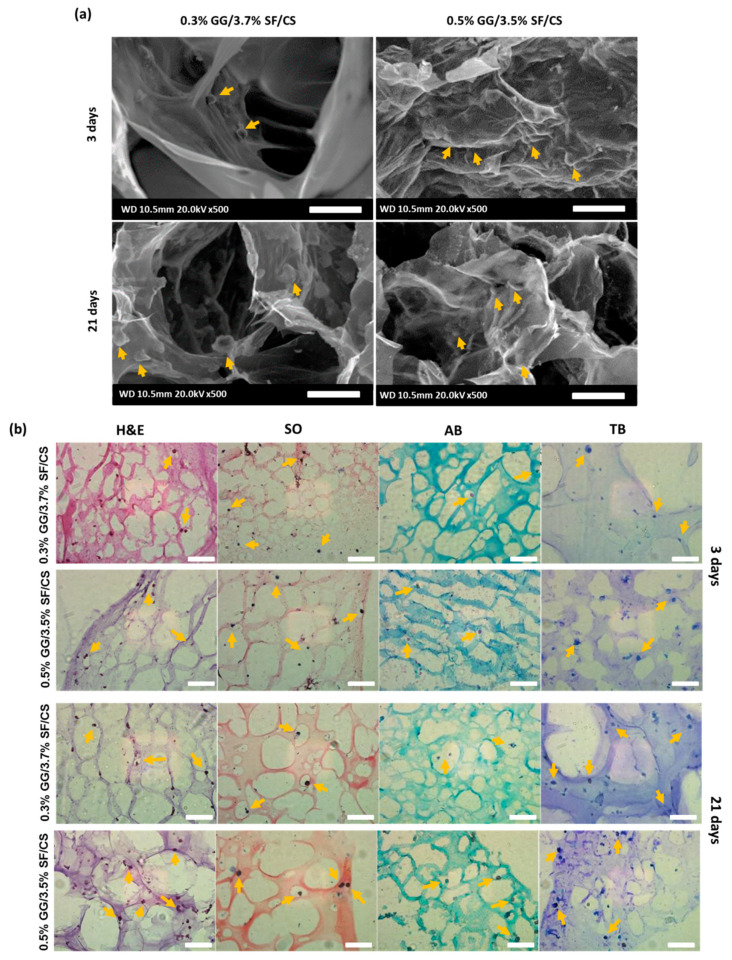
(**a**) Morphological observation of the cell-laden hydrogels under SEM analyzed on 3 and 21 days of culture. The cells were observed on 21 days of culture in both groups, which may be due to the proliferation of cells in the matrix (scale bar = 50 μm), and (**b**) histological analysis (H&E, SO, AB, and TB) was observed with the cell-laden hydrogels cells cultured in the ternary hydrogels for 3 and 21 days. On 21 days, the cell attachment in hydrogels occurred well, and cell proliferation occurred (scale bar = 25 μm).

**Figure 8 biomolecules-11-01184-f008:**
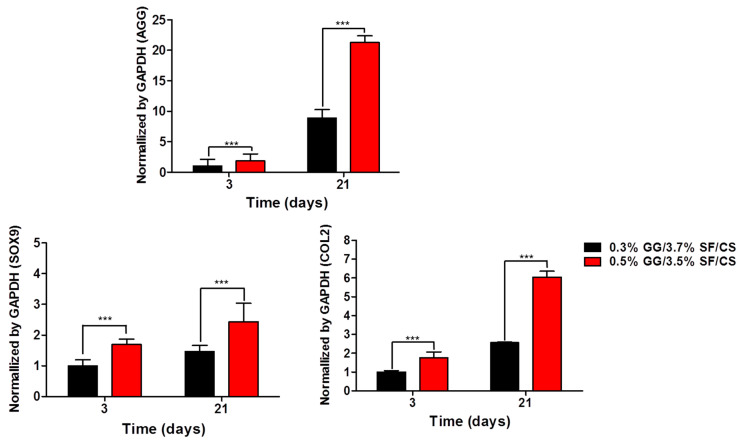
Cartilage-specific gene expression evaluated by RT-PCR with AGG, SOX9, and COL2 normalized by GAPDH (values are mean ± SD, *n* = 3, *p* < 0.001 (***)).

## Data Availability

The data presented in this study are openly available in [repository name e.g., FigShare] at [doi], reference number [reference number].

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
