# Peer review of "Development and Evaluation of Gellan Gum/Silk Fibroin/Chondroitin Sulfate Ternary Injectable Hydrogel for Cartilage Tissue Engineering"

_biomolecules, 2021, doi:10.3390/biom11081184_

Round 1

Reviewer 1 Report

Title: Development and evaluation of gellan gum/silk fibroin/chon-droitin sulfate ternary injectable hydrogel for cartilage tissue engineering

Objective: To develop a hydrogel of CS, GG and SF and evaluate its potential for cartilage tissue engineering.

Key message: The hydrogel composed of 0.5% GG/3.5% SF/CS is the most promising for cartilage tissue engineering, of all the hydrogels tested.

General comments and significance of the paper: Developing new and improved matrices for cartilage tissue engineering remains relevant. The authors prepared several hydrogels composed of GG, SF and CS and determined which one holds the most promise for cartilage tissue engineering applications. The study was well executed and the reviewer has only a few minor comments.

Specific comments:

  • The manuscript should be further edited, there are some typos and a few syntax errors throughout.
  • Page 1: Is the word “zinc” correct?
  • Page 2: Do you mean random coils?
  • Page 5: Was ascorbic acid supplementation used for the culture of chondrocytes?
  • Page 6: Can you add the genes + housekeeping gene in the methods?
  • Page 8: Keep only significant digits for pore size measurements.
  • Page 9: Red and blue arrows are missing in Figure 4.
  • Page 12: It seems that the DNA quantification does not support that proliferation was increased in both groups?
  • Page 12: Do you mean noticeable matrix around the cells?
  • Page 13: You did not compare with hydrogels containing only one or two of the components, can you truly say that the physicochemical and mechanical properties of hydrogels have improved with the integration of GG/SF/CS?

Author Response

Reviewer 1

We gratefully acknowledge the referees’ suggestions and criticisms, which can certainly improve the significance and quality of our manuscript. In the following, we answered all the questions and comments, and included all in the revised version of our manuscript. The revised parts are all marked in red.

Comment 1.Page 1: Is the word “zinc” correct?

-Thank you for your point out. We deleted the zinc that had nothing to do with the content. The revised paragraphs are in 1. Introduction and line 7.

Comment 2.Page 2: Do you mean random coils?

-Thank you for your point out. We changed arbitrary coil to random coil. The revised paragraphs are in 1. Introduction and line 51.

Comment 3.Page 5: Was ascorbic acid supplementation used for the culture of chondrocytes?

-Thank you for your detailed point out. Ascorbic acid supplementation was not applied in this study as the existence of ascorbic acid does not have significant difference in chondrogenic. (Asnaghi MA, Duhr R, Quasnichka H, Hollander AP, Kafienah W, Martin I, Wendt D. Chondrogenic differentiation of human chondrocytes cultured in the absence of ascorbic acid. J Tissue Eng Regen Med. 2018 Jun;12(6):1402-1411. doi: 10.1002/term.2671. Epub 2018 May 16. PMID: 29726103.)

Comment 4.Page 6: Can you add the genes + housekeeping gene in the methods?

-Thank you for your suggestion. We added the genes + housekeeping gene in the methods. The revised paragraphs are in 2.6.6. Real-time polymerase chain reaction (RT-PCR) and line 11.

Comment 5.Page 8: Keep only significant digits for pore size measurements.

-Thank you for your point out. We changed the significant digits for pore size measurements. The revised paragraphs are in 3.1.2. Morphological observation of hydrogel and line 3.

Comment 6.Page 9: Red and blue arrows are missing in Figure 4.

-Thank you for your detailed point out. We added the red and blue arrows in Figure 4 of the revised manuscript.

Comment 7.Page 12: It seems that the DNA quantification does not support that proliferation was increased in both groups?

-Thank you for your suggestion. We revised poor discussion and the revised paragraphs are in 3.2.2. dsDNA and GAG and line 3-19.

Comment 8.Page 12: Do you mean noticeable matrix around the cells?

-Thank you for your detailed point out. We changed cytoplasm to matrix. The revised paragraphs are in 3.2.3. Morphology and histological observation and line 8.

Comment 9.Page 13: You did not compare with hydrogels containing only one or two of the components, can you truly say that the physicochemical and mechanical properties of hydrogels have improved with the integration of GG/SF/CS?

-We gratefully acknowledge the reviewer’s comment. First, we agree that it is important to compare the ternary hydrogels with the hydrogels containing only one or two of the components. Considering that CS is constant in all the groups, we believe that it is best to compare with pristine GG or pristine SF. However, it is difficult to seek the optimal concentration to compare with the ternary hydrogels. This is because, pristine GG has poor mechanical property unless the concentration is above 1.5% or higher amount of calcium chloride is added for the ionic crosslinking. SF also cannot form hydrogel unless external stimulus (sonication, high temperature, etc.) is applied. Such differences are not considered suitable for use as a control group. Therefore, the best way was to compare with the previous studies that has similar property with the biomaterial we designed. We revised result and discussion of the physcicochemical study. The revised paragraphs are in 3.1.3. Physicochemical study of GG/SF/CS in line 2-9 and 12-13 and 3.1.4. Mechanical properties characterization line 3-10 of the revised manuscript. In the process of revision, stress and initial elastic modulus were calculated incorrectly, so this part was also corrected.

Again, we gratefully acknowledge the referees’ suggestions and criticisms and we hope the overall respond is satisfying to the reviewer.

Reviewer 2 Report

This paper is a report on fabrication and characterization of an injectable hydrogel scaffold for cartilage tissue engineering. I believe this work needs significant improvement to be ready for publication and here I discussed some of my concerns:

  • The manuscript needs English proof-reading.
  • There is a tremendous progress in commercialised solutions for cartilage tissue engineering and huge amount of work under research regarding fabrication and characterization of injectable formula. Authors ignored these current results and did not mention the state-of-the-art and challenges of the field. The motivation of investigating “ternary composite” was not explained. This needs to be changed in the introduction section.
  • Rationales behind using this composite with these specific ratios are not clear.
  • No control material has been studied to show how well or worse this ternary composite performs as an injectable scaffold for cartilage tissue engineering. For example, a generic material (e.g., collagen) should be studied along with this hydrogel to clarify of the advantages, or similar results should be cited from other references and compared with results of this work.
  • “Results and discussion” is basically only “results” and there is no discussion. Other similar works should be cited, discussed and compared to put the whole idea into context.
  • Authors reported several techniques, but they did not put any interpretation. For example, there is no explanation why cells are more active in one composite but not the other.
  • Figure 3. How do you interpret negative rate of degradation?
  • Figure 8. Objects that are identified as chondrocytes do not clearly express the morphology of these cells. More magnification and contrast are necessary to use these images as evidence.
  • It is not explained or hypothesized why an slight change in the ratio of GG/SF/CS results in a huge difference in gene expression and cell support? These results should be compared with standard material (e.g. collagen) to be meaningful.

Author Response

Reviewer 2

We gratefully acknowledge the referees’ suggestions and criticisms, which can certainly improve the significance and quality of our manuscript. In the following, we answered all the questions and comments, and included all in the revised version of our manuscript.

This paper is a report on fabrication and characterization of an injectable hydrogel scaffold for cartilage tissue engineering. I believe this work needs significant improvement to be ready for publication and here I discussed some of my concerns:

Comment 1.There is a tremendous progress in commercialized solutions for cartilage tissue engineering and huge amount of work under research regarding fabrication and characterization of injectable formula. Authors ignored these current results and did not mention the state-of-the-art and challenges of the field.

-Thank you for your detailed point out. We added a tremendous progress in commercialized solutions for cartilage tissue engineering and huge amount of work under research regarding fabrication and characterization of injectable formula. The revised paragraphs are in 1. Introduction and line 40.

Comment 2.The motivation of investigating “ternary composite” was not explained. This needs to be changed in the introduction section.

-Thank you for your point out. In this paper, three biomaterials were applied to design a hydrogel that simulate cartilage environment. Chondroitin sulfate, gellan gum, and silk fibroin, which are widely used in cartilage tissue engineering were used in this study. Each has its own advantages (chondroitin sulfate – provide an environment similar to cartilage tissue/ gellan gum – provides an environment and physical properties to cartilage tissue/ silk fibroin – has high mechanical property, support cell attachment and migration). However, in physical and biochemical point of view, only applying only one biomaterial has limitations to use in cartilage tissue engineering. Therefore, ternary hydrogel may complement the shortcomings of each biomaterial and integrate the merits. This is explained in the last 1. Introduction line 20-42 of the revised manuscript.

Comment 3. Rationales behind using this composite with these specific ratios are not clear.

- Thank you for your comment and criticism. When determining the proportion of composite, chondroitin sulfate was fixed to 1.5% based on reported studies; Yan, S., Zhang, Q., Wang, J., Liu, Y., Lu, S., Li, M., & Kaplan, D. L. (2013). Silk fibroin/chondroitin sulfate/hyaluronic acid ternary scaffolds for dermal tissue reconstruction. Acta Biomaterialia, 9(6), 6771–6782./ Zhou, F., Zhang, X., Cai, D., Li, J., Mu, Q., Zhang, W., … Ouyang, H. W. (2017). Silk fibroin-chondroitin sulfate scaffold with immuno-inhibition property for articular cartilage repair. Acta Biomaterialia, 63, 64–75. doi:10.1016/j.actbio.2017.09.005/ Ko, C.-S., Huang, J.-P., Huang, C.-W., & Chu, I.-M. (2009). Type II collagen-chondroitin sulfate-hyaluronan scaffold cross-linked by genipin for cartilage tissue engineering. Journal of Bioscience and Bioengineering, 107(2), 177–182. doi:10.1016/j.jbiosc.2008.09.020. The ratio of gellan gum and chemically crosslinked silk was chosen based on early experiment (not reported). Three factors were taken into account to choose the proper ratio. First, the concentration of gellan gum should not result in too high gelation property of the composite. Second, the silk must be well dispersed in the gellan gum solution. Third, the overall concentration should be constant. An initial experiment showed that when the concentration of gellan gum was above 1% in the ternary composite, the gelation occurred too quickly and at high temperature (>45 ℃). Therefore, gellan gum with a concentration below 1% was applied in the study. When chemically crosslinked silk fibroin with a concentration above 4% was blended with gellan gum, impurities such as fibroin appeared and gelation temperature of the composite also increased. Therefore, the ratio of gellan gum and silk was properly adjusted considering these conditions. We hope our response is satisfying to the reviewer.

Comment 4. No control material has been studied to show how well or worse this ternary composite performs as an injectable scaffold for cartilage tissue engineering. For example, a generic material (e.g., collagen) should be studied along with this hydrogel to clarify of the advantages, or similar results should be cited from other references and compared with results of this work.

- We gratefully acknowledge the reviewer’s comment. First, we fully agree that it is important to set a control to compare the ternary hydrogels. Considering that CS is constant in all the groups, we believe that it is best to compare with pristine GG or pristine SF. However, it is difficult to seek the optimal concentration to compare with the ternary hydrogels. This is because, pristine GG has poor mechanical property unless the concentration is above 1.5% or higher amount of calcium chloride is added for the ionic crosslinking. SF also cannot form hydrogel unless external stimulus (sonication, high temperature, etc.) is applied. Such differences are not considered suitable for use as a control group. Therefore, the best way was to compare with the previous studies that has similar property with the biomaterial we designed. We revised result and discussion of the physcicochemical study. The revised paragraphs are in 3.1.3. Physicochemical study of GG/SF/CS in line 2-9 and 12-13 and 3.1.4. Mechanical properties characterization line 3-10 of the revised manuscript. We did not consider applying a generic material for the control as we were to find the effective ratio of the gellan gum, silk fibroin, and chondroitin sulfate and evaluate the difference among them. We will fully consider using generic material as a control in the future study. We hope our response is satisfying to the reviewer.

Comment 5.“Results and discussion” is basically only “results” and there is no discussion. Other similar works should be cited, discussed and compared to put the whole idea into context.

-Thank you for your detailed point out. We revised overall results and discussion which are all marked in red.

Comment 6.Authors reported several techniques, but they did not put any interpretation. For example, there is no explanation why cells are more active in one composite but not the other.

-Thank you for your detailed point out. We revised overall results and discussion which are all marked in red.

Comment 7. Figure 3. How do you interpret negative rate of degradation?

-We gratefully acknowledge the reviewer’s point out. The reason for the negative rate of degradation was that swelling occurred for 24 hrs.

Comment 8. Figure 8. Objects that are identified as chondrocytes do not clearly express the morphology of these cells. More magnification and contrast are necessary to use these images as evidence.

-Thank you for your point out. We enlarged the image in Figure 8 and adjusted the contrast as the shape of the cells was not clear.

Comment 9.It is not explained or hypothesized why a slight change in the ratio of GG/SF/CS results in a huge difference in gene expression and cell support? These results should be compared with standard material (e.g. collagen) to be meaningful.

-Thank you for your advice. We added more explanation and hypothesis of why different ratio of GG/SF/CS results in a huge difference in gene expression and cell support. The revised paragraphs are in the 3.2.4. Gene expression line 5-9 of the revised manuscript. As it is shown in the mechanical property, higher ratio of GG significantly enhances the mechanical property of the hydrogel and it was suspected that this enhanced mechanical property of the hydrogel may have sufficiently supported the phenotype of the chondrocytyes. We fully agree that these results should be compared with the standard material. However, it was difficult to compare with standard material as the experimental conditions were different (degradation rate, mechanical property, and fabrication method). We did not consider applying a generic material for the control as we were to find the effective ratio of the gellan gum, silk fibroin, and chondroitin sulfate and evaluate the difference among them. In the future study. we will fully consider using standard material as a control and compare with the 0.5% GG/3.5% SF/CS which showed the best result. in the future study. We hope our response is satisfying to the reviewer.

Again, we gratefully acknowledge the referees’ suggestions and criticisms and we hope the overall respond is satisfying to the reviewer.

Reviewer 3 Report

The manuscript entitled “Development and evaluation of gellan gum/silk fibroin/chondroitin sulfate ternary injectable hydrogel for cartilage tissue engineering” by Seong Won Lee described the formulation and characterization of ternary-hydrogels for cartilage tissue engineering. In detail hydrogels are based on three different natural polymers (gellan gum, silk fibroin and chondroitin sulfate) at different concentration. The physic-chemical and mechanical properties were evaluated and biological response were tested.

Althought authors presented good results, some revisions have to be done before publication in Biomolecules.

  • Abstract. Abstract should be revised in order to better introduce and summarize the scientific work reported in the manuscript. Please add a first sentence to introduce the application field of the hydrolgels.

The sentences “The hydrogels were fabricated by modifying the GG and SF concentration and adjusting the CS. The physicochemical properties were characterized to analyze porous structure, water uptake, crosslinking density, and decomposition rate. The mechanical characters were measured to analyze the compression modulus and gelation properties of the hydrogels” are not clear, please revised the description and charcaterization for the hydrogel fabrication.

  • Introduction. The first part of the introduction, describing cartilage tissue and related pathologies, is not clear and should be implemented and focused in order to better introduce the scientific work reported in the manuscript. Authors do not report the state of the art about hydrogels used in cartilage application and do not highlight the innovation of their work, please add these important information.
  • Material and Methods:
    • Why the same calcium chloride concentration was used for all the developed hydrogels?
    • Please add the sample acronyms in the followed sentence: “The crosslinked SF solution was included in each GG/CS solution to make total concentration of 3.7%, 3.5%, 3.25%, and 3% of SF in 0.3% GG/CS, 0.5% GG/CS, 0.75% GG/CS, and 1% GG/CS, respectively.”
    • Please add more information about infrared spectroscopy: resolution, scan number and crystal type.
    • The swelling analysis should be carried out up to 24h, that the time used for the degradation test.
    • Degradation test and sol fraction should be performed at the same condition. I suggest to authors to carried out these two analysis in both PBS and enzymatically condition at the same time steps.
  • Results:
    • Authors should add to Figure 1 the FTIR-ATR spectra of each hydrogel.
    • Figure 2: please enlarge the Figure in order to better highlight Fiure 2b
    • Figure 3: the sem images of Figure 8a should be enlarged.
    • In general the results discussion is too rough, Authors should better discuss the obtained results referring to the literature state of the art.

Author Response

Reviewer 3

We gratefully acknowledge the referees’ suggestions and criticisms, which can certainly improve the significance and quality of our manuscript. In the following, we answered all the questions and comments, and included all in the revised version of our manuscript.

The manuscript entitled “Development and evaluation of gellan gum/silk fibroin/chondroitin sulfate ternary injectable hydrogel for cartilage tissue engineering” by Seong Won Lee described the formulation and characterization of ternary-hydrogels for cartilage tissue engineering. In detail hydrogels are based on three different natural polymers (gellan gum, silk fibroin and chondroitin sulfate) at different concentration. The physic-chemical and mechanical properties were evaluated and biological responses were tested.

Although authors presented good results, some revisions have to be done before publication in Biomolecules.

Comment 1.Abstract should be revised in order to better introduce and summarize the scientific work reported in the manuscript. Please add a first sentence to introduce the application field of the hydrogels.

-Thank you for your opinion. We revised in order to better introduce and summarize the scientific work reported in the manuscript. The revised paragraphs are in Abstract and line 1-4 of the revised manuscript.

Comment 2. The sentences “The hydrogels were fabricated by modifying the GG and SF concentration and adjusting the CS. The physicochemical properties were characterized to analyze porous structure, water uptake, crosslinking density, and decomposition rate. The mechanical characters were measured to analyze the compression modulus and gelation properties of the hydrogels” are not clear, please revised the description and charcaterization for the hydrogel fabrication.

-Thank you for your comment. We revised the description and characterization for the hydrogel fabrication. The revised paragraphs are in Abstract and line 5-6.

Comment 3.The first part of the introduction, describing cartilage tissue and related pathologies, is not clear and should be implemented and focused in order to better introduce the scientific work reported in the manuscript. Authors do not report the state of the art about hydrogels used in cartilage application and do not highlight the innovation of their work, please add these important information.

-Thank you for your point out. We revised overall introduction and the revised parts are all marked in red.

Comment 4.Why the same calcium chloride concentration was used for all the developed hydrogels?

-Thank you for your point out. Several variables arise from controlling the concentration of GG and SF. If the ion concentration is also adjusted, the variables become to numerous. Therefore, we author thought that the amount of ion should be fixed and only the amount of the GG and SF should be controlled. However, since physical properties and gelation rate and temperature are influenced by the ion content, we will consider it in the future studies.

Comment 5.Please add the sample acronyms in the followed sentence: “The crosslinked SF solution was included in each GG/CS solution to make total concentration of 3.7%, 3.5%, 3.25%, and 3% of SF in 0.3% GG/CS, 0.5% GG/CS, 0.75% GG/CS, and 1% GG/CS, respectively.”

-Thank you for your point out. We added the sample acronyms in the followed sentence. The revised paragraphs are in 2.2. Fabrication of hydrogels and line 8-9.

Comment 6.Please added more information about infrared spectroscopy: resolution, scan number and crystal type.

-Thank you for your opinion. We added more information about infrared spectroscopy: resolution and crystal type. Unfortunately, we could not find the scan number. We hope included information is satisfying to the reviewer. The revised paragraphs are in 2.3. Fourier-transform infrared (FT-IR) spectroscopy and line 2-3.

Comment 7. The swelling analysis should be carried out up to 24h, that the time used for the degradation test.

-Thank you for your point out. We changed 72 hrs to 24 hrs in Fig. 3 (a). The revised paragraphs are in 3.1.3. Physicochemical of GG/SF/CS and line 4.

Comment 8.Degradation test and sol fraction should be performed at the same condition. I suggest to authors to carried out these two analysis in both PBS and enzymatically condition at the same time steps.

-We gratefully acknowledge the reviewer’s comment. However, sol fraction of the hydrogel represents the fraction of the polymer that is not involved in the crosslinked network which show the crosslinking density of the polymer matrix (Li et al., W. Lee et al., M. Suhail et al.). In order to analyze it, most of the papers evaluate it under a good solvent of the material. On the other hand, degradation test depends on various physical factors such as crosslinking density, crystallinity, etc. and the test is performed to analyze the change in composite overtime under physiological condition and to ensure that the composite can provide the necessary support for cells to regenerate completely. Therefore, we author believe that it is unnecessary to measure the sol fraction in enzymatic conditions because the two observation points are different.

Comment 9.Authors should add to Figure 1 the FTIR-ATR spectra of each hydrogel.

-Thank you for your point out. We added the FTIR-ATR spectra of each hydrogel in Fig. 1. The revised paragraphs are in 3.1.1. FT-IR analysis and line 6-9.

Comment 10.Figure 2: please enlarge the Figure in order to better highlight Figure 2b.

-Thank you for your point out. We enlarge the Figure in order to better highlight in Fig. 2 (b).

Comment 11.Figure 3: the sem images of Figure 8a should be enlarged.

-Thank you for your point out. We enlarged the sem images of Fig. 8 (a).

Comment 12.In general the results discussion is too rough, Authors should better discuss the obtained results referring to the literature state of the art.

-Thank you for your detailed point out. We revised overall results and discussion which are all marked in red.

Again, we gratefully acknowledge the referees’ suggestions and criticisms and we hope the overall respond is satisfying to the reviewer.

Round 2

Reviewer 3 Report

Authors revised the manuscript according the comments. I recommend the manuscript in the present form for biomolecules journal.